# Revisiting Current Trends in Electrode Assembly and Characterization Methodologies for Biofilm Applications

**Luis Alberto Estudillo-Wong [1], Claudia Guerrero-Barajas [2,\*], Jorge Vázquez-Arenas [3] and Nicolas Alonso-Vante [4,\*]**

1 Departamento de Biociencias e Ingeniería, CIIEMAD, Instituto Politécnico Nacional, Calle 30 de junio de 1520 s/n, Barrio la Laguna Ticomán, Alcaldía GAM, Mexico City 07340, Mexico

2 Departamento de Bioprocesos, Laboratorio de Biotecnología Ambiental, Unidad Profesional Interdisciplinaria de Biotecnología, Instituto Politécnico Nacional, Av. Acueducto s/n, Col. Barrio la Laguna Ticomán, Alcaldía GAM, Mexico City 07340, Mexico

3 Centro Mexicano para la Producción más Limpia, Instituto Politécnico Nacional, Av. Acueducto s/n, Col. la Laguna Ticomán, Alcaldía GAM, Mexico City 07340, Mexico

4 Institut de Chimie des Milieux et Matériaux de Poitiers (IC2MP), UMR-CNRS 7285, University of Poitiers, F-86073 Poitiers, France

\* Correspondence: cguerrerob@ipn.com (C.G.-B.); nicolas.alonso.vante@univ-poitiers.fr (N.A.-V.)

**Abstract:** Microbial fuel cell (MFC) is a sustainable technology resulting from the synergism between biotechnology and electrochemistry, exploiting diverse fundamental aspects for the development of numerous applications, including wastewater treatment and energy production. Nevertheless, these devices currently present several limitations and operational restrictions associated with their performance, efficiency, durability, cost, and competitiveness against other technologies. Accordingly, the synthesis of nD nanomaterials (n = 0, 1, 2, and 3) of particular interest in MFCs, methods of assembling a biofilm-based electrode material, in situ and ex situ physicochemical characterizations, electrochemistry of materials, and phenomena controlling electron transfer mechanisms are critically revisited in order to identify the steps that determine the rate of electron transfer, while exploiting novel materials that enhance the interaction that arises between microorganisms and electrodes. This is expected to pave the way for the consolidation of this technology on a large scale to access untapped markets.

**Keywords:** microbial fuel cell; biofilm; wastewater; energy production; assembly methods

## 1. Introduction

Microbial fuel cell (MFC) is a promising bio-electrochemical technology with the capacity to degrade organic matter while producing electricity, whereby it constitutes a sustainable alternative in environmental science [1]. In MFCs, residual organic biomass or wastewater is biologically oxidized by exoelectrogenic bacteria, where direct or indirect electron transfer occurs between the microorganism and the terminal electron acceptor [1,2]. In most cases, the terminal electron acceptor is an anode electrode, and the electrons are transported through an electronic conductor to the cathode to perform the Oxygen Reduction Reaction (ORR). However, the ORR sluggish kinetics in most materials is a disadvantage in MFCs [3], as scaling is limited to improve the electrocatalytic activity of the cathode, in addition to using expensive materials. In this context, several research groups have been working to overcome these drawbacks in MFCs to consolidate this technology at the commercial level [4–8]. To this concern, non-precious metals are currently investigated as a novel alternative to replace the platinum group metals (PGMs) [9]. In addition, synthesis, stability, methanol tolerance, and selectivity of the material are real challenges to improve the performance of MFCs [10,11].

On the contrary, anode materials are of great importance for electricity generation because microorganisms adhere to them and grow on them. This process is directly related to

the metabolic activity of the bacteria; thus, the anode material should be biocompatible [12]. Under this premise, electrode preparation methods play a crucial role in increasing the contact surface area. Different types of nanomaterials are synthetized using top-down and bottom-up approaches, and employing physical and chemical methods to prepare 0D, 1D, 2D, and 3D nanomaterials. The 0D nanomaterials are often involved in antibacterial activity [13,14]. The 0D and 1D nanomaterials can be synthesized as composites to accelerate the biofilm formation and facilitate direct electron transfer [15]. In the case of 2D morphology, these nanomaterials are used as support in combination with 0D and 3D nanomaterials to test biocompatibility [16] or activity towards ORR reaction [17,18]. Meanwhile, to increase the large specific area for microbial adhesion, 3D nanomaterials are formed between the cathode and anode by filling the inner part with 0D nanomaterials [19], which greatly improves mass transfer. In order to rationalize the underlying chemistry controlling the overall performance of MFCs, their electrocatalytic effects and current efficiency, physico-chemical and electrochemical characterizations of the nanomaterials are performed under in situ [20–28] and ex situ [29–33] modes. In ex situ analysis, High-Resolution Transmission and Scanning Electron Microscopies (HRTEM and SEM), X-ray Diffraction (XRD), Atomic Force Microscopy (AFM), and X-ray photoelectron spectroscopy (XPS) are commonly used to characterize the structural and textural properties of the nanomaterials used in the cells. Other techniques, such as Electrochemical on-line Inductively Coupled Plasma Mass Spectroscopy (ICP-MS) [25] and Differential Electrochemical Mass Spectroscopy (DEMS) [26], can be implemented to perform a comprehensive transient evaluation of the electrochemical system during operation, thus, revealing short-time phenomena, complex interface features, and intermediates products and providing structure–activity–signal relationship information. These results provide a complete picture of the by-products formed and offer insight into the interaction between nanomaterials and biofilms, where the electroactivity of the MFCs was enhanced.

Currently, wastewater is considered a renewable resource of energy [34]. In this case, MFCs can serve as a link to treat the unit wastewater while producing energy. Therefore, efforts are being directed towards developing this bio-electrochemical technology and improving sanitation and water quality, as pointed out in goal six of the United Nation Sustainable Development Goals (SDG) [35].

In this feature article, we address the synthesis of nD nanomaterials (n = 0, 1, 2, and 3), their physicochemical and electrochemical characterizations with in situ (i.e., during data acquisition or experimental performance) and ex situ (i.e., after experimental collection) approaches, and their assembly within a biofilm-based electrode material. Finally, we include the electron transfer mechanism to understand the behavior of MFC applied to wastewater treatment.

## 2. Electrode Preparation Methods

Methods for 0D to 3D nanomaterials. Materials Science defines nanomaterials as advanced materials, where the dimensions of these structural entities are in the order of a nanometer and less than 100 nanometers. Figure 1 shows the classification of nanomaterials, in which the 3D dimension changes from [000] to [xyz] directions.

Different synthesis methods, such as physical and chemical methods [36], can be applied to prepare 0D, 1D, 2D, and 3D nanomaterials using top-down and bottom-up approaches. Hydrothermal, solvothermal [37–39], sol-gel [5,40], and chemical vapor deposition (CVD) methods [41,42] are commonly used to synthesize 0D nanomaterials. The latter have some advantages and limitations, including the possibility of integrating them into multiple systems and agglomerates [43]. Figure 2 shows a common procedure to synthesize 0D nanoparticles by hydrothermal, sol-gel, and vapor chemical deposition routes [44].

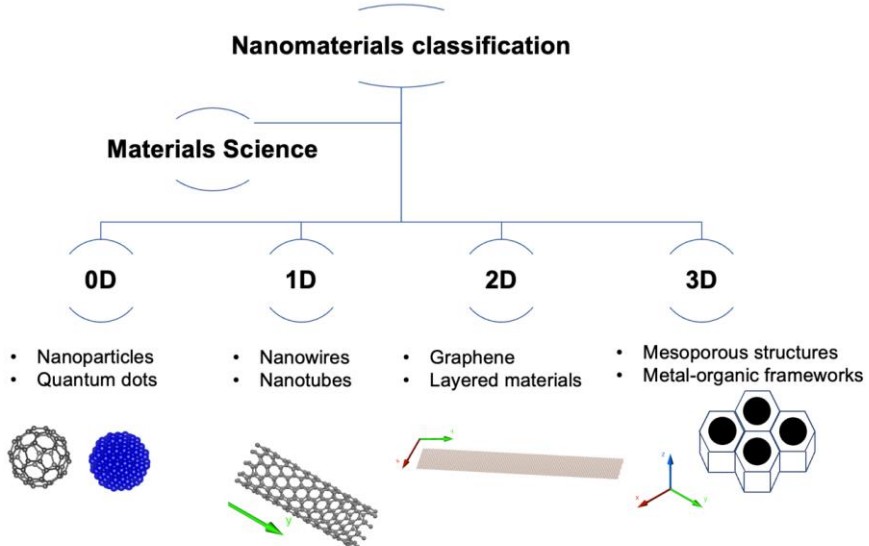

**Figure 1.** Classification of nD nanomaterials (n = 0, 1, 2, and 3).

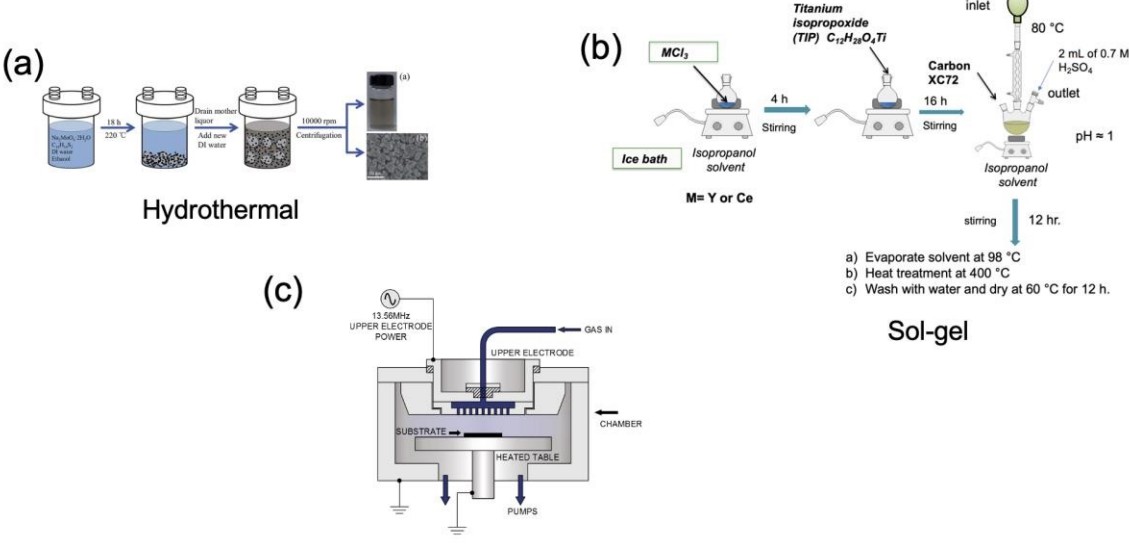

**Figure 2.** Synthesis methods for the preparation of nD nanomaterials (n = 0, 1, 2, and 3): (**a**) Hydrothermal, with permission from Ref. [39]. Copyright (2015) Royal Society of Chemistry. (**b**) Solgel; and (**c**) Chemical Vapor Deposition, from Ref. [45].

For 1D nanomaterials, electrochemical deposition [46], electrospinning [47,48], and chemical vapor deposition [46] are used to synthetize nanotubes, nanorods, and nanowires objects. The advantages involve the possibility to integrate them in low-cost devices, and their limitations are low production and high cost. Figure 3 illustrates the electrodeposition and electrospinning methods.

In the case of 2D nanomaterials, Ruan et al. [49] revised different synthesis methods and classified them, as described in Figure 4. Here, we can observe that chemical vapor deposition and hydrothermal-solvothermal methods are the most common methods. Some advantages are (i) open 2D channels for ion transport and (ii) compatible with flexible devices, while limitations involve the high synthesis cost.

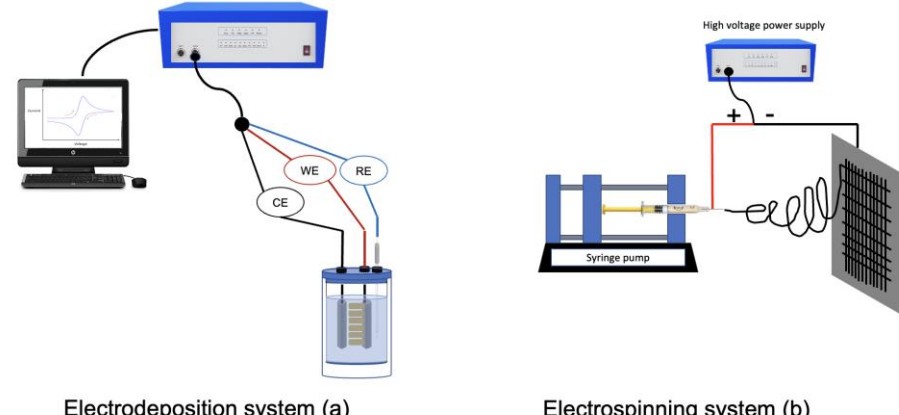

**Figure 3.** (**a**) Electrodeposition, and (**b**) Electrospinning synthesis methods applied for 1D nanomaterials.

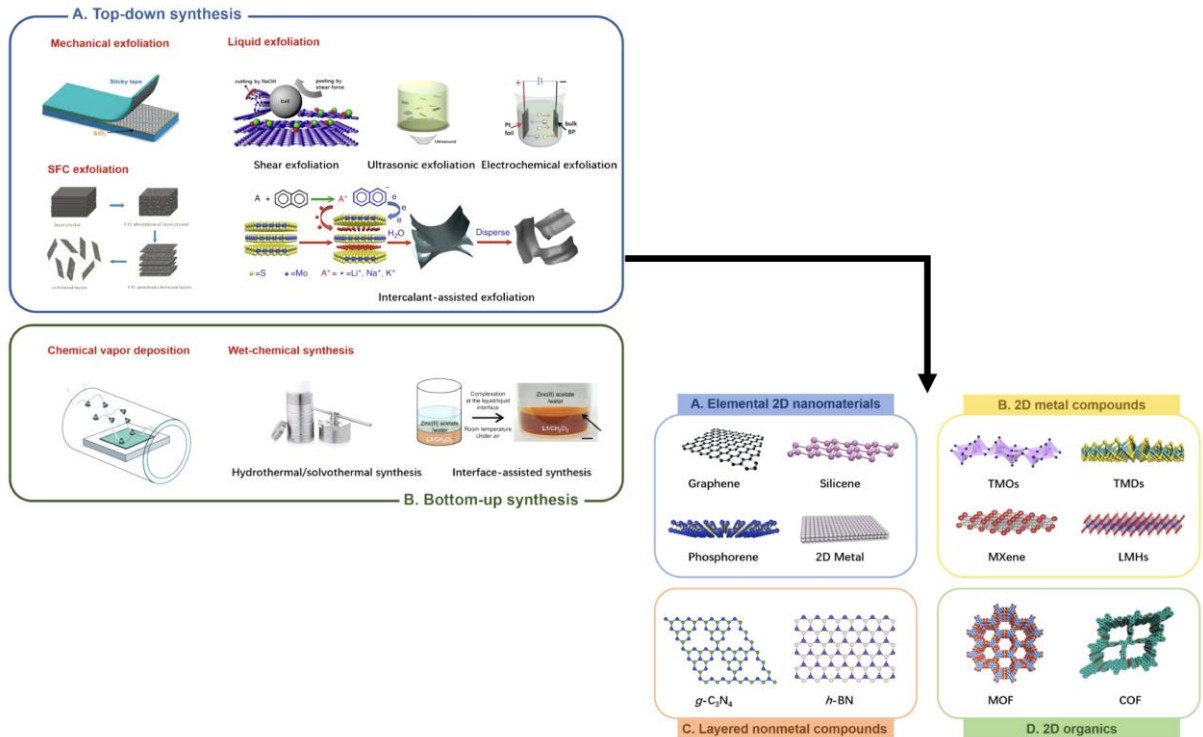

**Figure 4.** Some strategies for the synthesis of 2D nanomaterials. Figure modified and adapted with permission from Ref. [49]. Copyright (2021) Elsevier.

In 3D morphology, Yu et. al. [50] reported that there are two categories, namely, (i) template synthesis and (ii) a template-free method, which were applied to synthesize $Fe_2O_3$ nanostructures. Mazzotta et al. [51] synthesized Pt flower-like nanostructures without a template. An additional interest is to produce 3D nanomaterials based on carbon nanomaterials, in which these materials are produced by bacterial cellulose from biomass [52]. Furthermore, transition metal compounds have been prepared, including $Co_3O_4$ [53], $TiO_2$ [54], and CoNiS [55], having a flower-like and dendritic morphologies. Therefore, we can find that different methods of synthesis can be applied to produce 3D nanomaterials, such as chemical vapor deposition [56], solvothermal-hydrothermal [53–55], electrodeposition [51], and pyrolysis [57]. Figure 5 shows the synthesis of CoNiS flower-like and the 3D morphology.

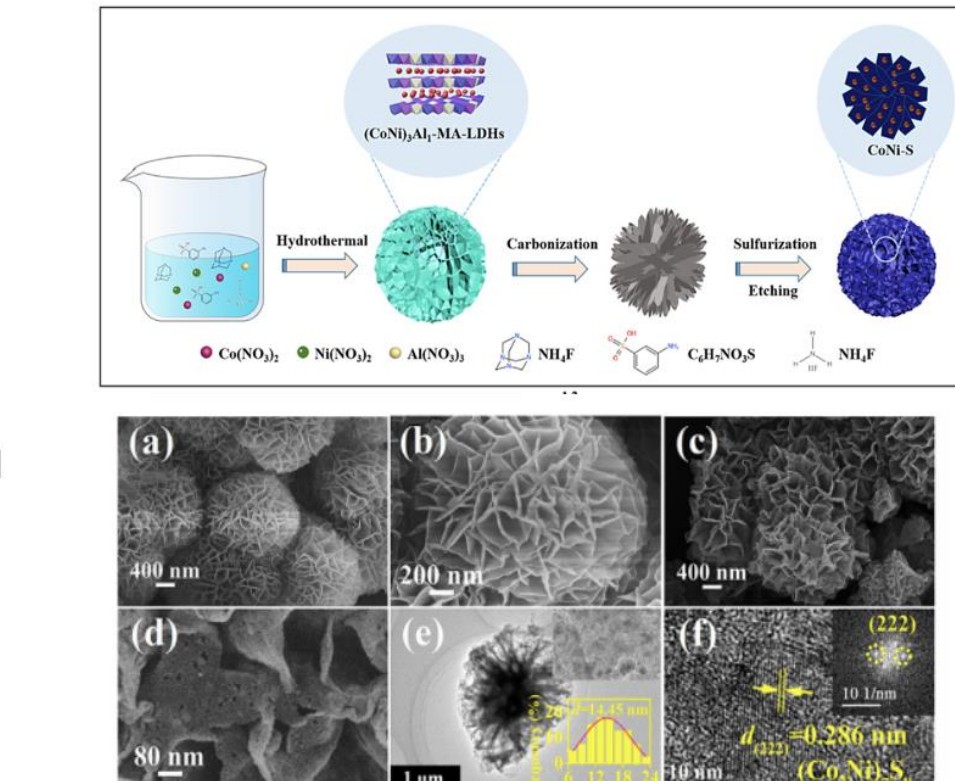

I

II

**Figure 5.** (**I**) Synthesis of 3D nanomaterials using non-precious metals, and (**II**) HRTEM analysis (**a**–**f**) for CoNiS samples. Figure modified and adapted with permission from Ref. [55]. Copyright (2022) Elsevier.

The major advantage of these materials is the surface area (>300 m$^2$ g$^{-1}$), while limitations entail their chemical stability. Finally, all these nD nanomaterials (n = 0, 1, 2, and 3) are being explored to create a biofilm with microorganisms, which are discussed in the following sections.

Currently, there is a renewed interest in developing sustainable technologies that offer energy savings or generation in relation to wastewater treatment. The variety of pollutants to be removed from wastewater is increasing, including emerging pollutants (non-biodegradable or photolytic decomposed), such as pharmaceutical compounds to treat new diseases, endocrine disruptors, organics derived from industrial activities, heavy metals, and toxic organic pollutants that threaten aquatic environments. In this context, bioelectrochemical systems (BES) have become a comprehensive area of study that require the combination of several disciplines to achieve their successful design and operation. MFCs are one of the most studied BES, which aim to supply energy in the form of electricity while removing pollutants from water (organic matter, specific chemicals, and metals). Despite all the efforts devoted to investigating the scaling of these devices in wastewater treatment, there are still some constraints to be faced, as the involvement of microorganisms in any process can be difficult to understand, especially in a large-scale and continuous regime.

The most common configuration of MFCs is depicted in Figure 6. Microorganisms and electrodes are of utmost importance, and therefore, research has focused on both the diversity of microorganisms that can act as biocatalysts in the reactions carried out in the MFCs and the electrodes that are used to support the microorganisms conglomerate or biofilm. As a result of extensive research, it has been possible to classify the microorganisms that have been used over the years in MFC as low electrogenic (power production < 10 mW m$^{-2}$), moderate (<100 mW m$^{-2}$), high (<1000 mW m$^{-2}$), and remarkably high (>1000 mW m$^{-2}$), depending on the power density obtained in these systems [58]. Despite this classification, the effectiveness of the microorganisms in supplying energy also depends on other factors, such as the characteristics of the biofilm (e.g., texture, hydrophobicity, roughness), its ability

to sustain the reactions occurring in the MFC chambers and electrodes (anodic or cathodic biofilms), and the type of material where the biofilm adheres. A cell may exhibit flagella, fimbriae, nanowires or pili, and excretion of extra polymeric substances (EPS) that serve during adhesion of the cell to the surface. A description of the forces involved during the initial stages of biofilm formation is detailed in references [59] and [60]. The main forces involve non-covalent Van der Waals or electrodynamic, Lewis's acid-base, and electrostatic (i.e., electrical double layer) interactions.

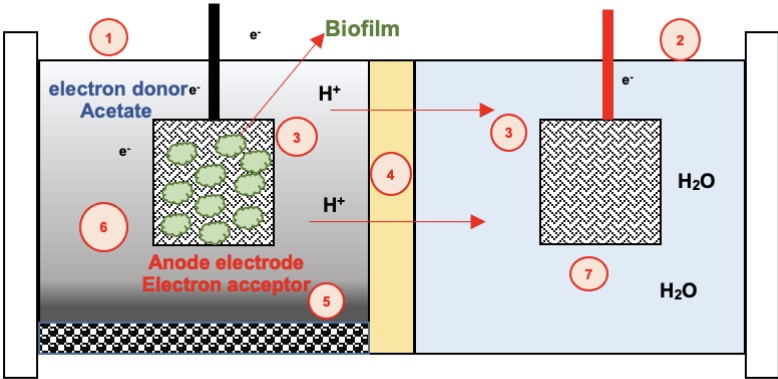

**Figure 6.** Example of a type "H" MFC. (1) Anodic chamber (anaerobic), (2) cathodic chamber (aerobic), (3) graphite cloth (electrodes), (4) cation exchange membrane, (5) electrogenic bacteria, (6) mineral medium, (7) distilled water. Redox reactions that occur in the MFC are as follows. anode reaction: $CH_3COO^-_{(aq)} + 4H_2O_{(l)} \rightarrow 2HCO^-_{3(aq)} + 9H^+_{(aq)} + 8e^-$; cathode reaction: $2O_{2(g)} + 8H^+_{(aq)} + 8e^- \rightarrow 4H_2O_{(l)}$.

The physical aspects related to the cells are the motility—which is given by the flagella assisting the cell movements through concentration gradients or chemotaxis—through electric fields or electro or Galvanotaxis, magnetic fields or magnetotaxis, and light or phototaxis [61]. As pointed out in a recent report, in which *Pseudomonas aeruginosa* was studied on the formation of biofilms, a description of biofilm formation should include the information on how the cell senses a surface and the important steps to construct a community on it [62]. In *Pseudomonas aeruginosa*, two surface sensing mechanisms have been identified; one of them is like a chemotaxis signal transduction network, with the involvement of methyl-accepting chemotaxis protein (MCP) and several other proteins, including methylase and methyl esterease. This mechanism assumes that the surface exerts certain stress on the cell membrane, and this triggers the protein production and development of the mechanism that involves the MCP [63]. The second surface sensing mechanism suggests that the pili are involved (Pil-Chp system), along with several intracellular molecules, and that the biofilm formation does not necessarily occur linearly in a progressive sequence of steps, but in successive surface interactions and detachments, through which cells finally become adapted to the surface [64]. In this mechanism, the Type IV pili, which are produced through the Pil-Chp system, are involved. When the pili encounters a surface, the MCP protein transduces signals to other molecules, leading to an increase of the cyclic AMP (cAMP). This latter is a second messenger signaling molecule that controls a wide variety of extracellular actions in cells [65]. It has also been discovered that the whole Pil-Chp surface system response does not occur in a scale of time of hours, but longer. Figure 7A presents the formation of biofilm in experiments conducted with *Pseudomonas aeruginosa*, according to Lee et al. [64]. It has been suggested that in mixed cultures, the first cells that attach to the surface create the binding sites for a second species to attach, and therefore, the formation of the biofilm is promoted [66]. From a more practical point of view, considering that the surface sensing mechanisms have occurred, the biofilm formation has been depicted according to Figure 7B, in which the various stages of formation are illustrated. These stages are described as follows: (a) the attachment,

reversible or irreversible; (b) the biofilm maturation; (c) the matrix formation with diffusion gradients; and (d) the erosion and different diffusion gradients.

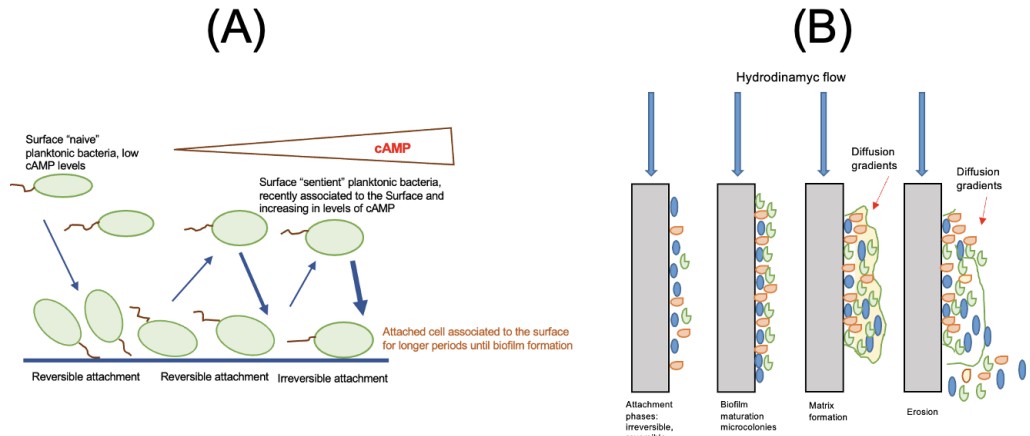

**Figure 7.** (**A**) Stages or biofilm formation. After successive attachment/detachment events, the cAMP levels gradually increase and build thorough the Pil-Chp surface sensing system. Adapted from Ref. [62]. Copyright (2018) PNAS. (**B**) Stages of a biofilm formation and disintegration, adapted from Ref. [66].

Ideally, in applications such as MFC, a biofilm serves as the primary catalyst for the process and is effective in transferring electrons to the anode due to the presence of electrogenic bacteria that exhibit several electron transfer mechanisms. In a consortium of bacteria, from which a biofilm has formed, more than one electron transfer mechanism may be expressed. Biofilms thicker than 20–50 mm have been reported to be diffusion limited. Therefore, a stratification occurs within these biofilms that implies an unequal distribution of substrate and electron acceptors for the microorganisms. In thick biofilms, the growth of the inner microorganisms is slower, and the biofilm tends to show erosion. It has also been claimed that, in addition to the surface sensing mechanisms, which are manifested to produce biofilms, the presence of extracellular polymeric substances or exopolysaccharides (EPS) and lipopolysaccharides (LPS) also plays a key role in biofilm production and electron transfer mechanisms when these biofilms are formed on anode surfaces. The role of the EPS in electron transfer was discovered through research on the electrogenic bacteria *Geobacter sulfurreducens*, which has genes encoding for exopolysaccharides that serve as binding sites for c-type cytochromes, a type of protein needed to transfer electrons to the electrode [67].

At least in the case of small-scale MFCs, it has been observed that the biofilm detachment stage is favored by the flow rate of the substrate (laminar flow) in the anaerobic chamber. This flow may serve as a shear force to remove cells that are weakly attached to the biofilm towards the planktonic bulk. It has been assumed that the adhesion forces are stronger for the cells that are directly attached to the electrode and form a "mother layer" than for cells that are in the middle of the biofilm. Therefore, the electrode becomes a "non -accumulator" site over time and retains its physicochemical properties, while the microbial growth rate will continue to depend on the flow rate in the chamber containing the substrate and chemical nutrients, until it reaches a "steady state" or a state close to steady state, in which the MFC could be operating [66].

### 3. Electron Transfer Mechanisms

In bioelectrochemical systems including MFCs, the role of the biocatalyst (biofilm) and its constituent microorganisms is crucial for the electrode performance, either bioanodes or biocathodes. Thus, it is important to understand the electron transfer mechanisms. This manuscript focuses on biofilms formed on anodes. The study of electroactive biofilms has been largely directed at the electron transfer mechanisms and the factors influencing them. The electron transfer mechanisms presented by electrogenic bacteria once they have formed a biofilm on the electrode surface can be direct (DET) or indirect (IDET), as described below.

Direct electron transfer (DET). In an MFC where the anode surface has been covered by an electroactive biofilm, Figure 8, a DET would imply that the electrons are transferred to the anode through "direct" contact with the bacteria. The bacteria can transfer electrons from their membrane to the electrode without any intermediate. In this case, bacteria membranes possess certain cytochrome-type proteins that act directly on electron transfer. Although this mechanism may be exhibited in mixed bacteria contained in the biofilm, it may involve highly electroactive bacteria other than the most studied ones, which has been widely confirmed in *Geobacter sulfurreducens* and *Rhodoferax ferrireducens*. At the same time, for the bacteria to adhere to the anode surface, EPS are expressed from the preliminary stages of the biofilm formation, and the first layer (a monolayer) of microorganisms growing on the anode surface would be the one involved in the DET. Therefore, it is important that this monolayer consists of highly electroactive bacteria. In addition to the outer-membrane cytochromes used in this electron transfer mechanism, electroactive bacteria can also express electroconductive pili or nanowires that allow reaching the surface of the anode and taking it as the sole electron acceptor. The conductive pili enables the DET through the creation of a denser biofilm due to the entanglement of the pili with other components of the biofilm (i.e., EPS). Additionally, a recent report [68] describes that the direct electron transfer between interspecies is a mechanism that adapts to changes in organic loads and in the presence of contaminants. This was observed during the oxidation of organic waste in anaerobic digestion, where bacteria could establish electrical connections with methane-producing microorganisms through protein-based conductive circuits. Direct interspecies electron transfer was observed to be faster than the interspecies electron exchange via diffusive electron carriers (or electron shuttles). This finding is, to the best of our knowledge, the most recent in relation to direct electron transfer and the role it may play in enhancing methane production in anaerobic digestion, which is one of the most relevant bioprocesses for the generation of energy.

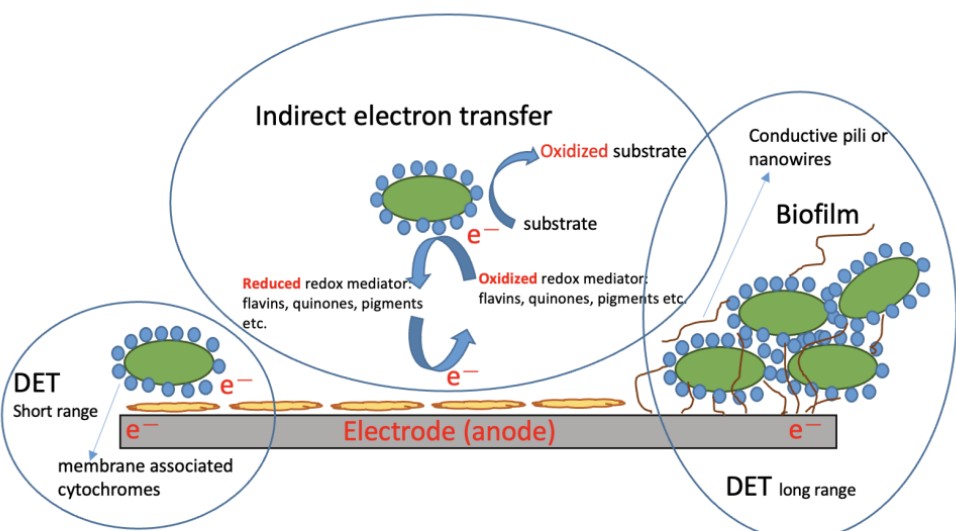

**Figure 8.** Electron transfer mechanism between biofilms and electrode materials.

Indirect electron transfer (IDET). Indirect electron transfer mechanisms are also known as mediated electron transfer (MET) mechanisms. In this type of process, a mediator known as a "redox mediator" or "electron shuttle" serves as a "reversible" electron acceptor, which reduces and then transfers the electrons to the anode surface, to a component of the medium, or to other bacteria in the biofilm. Thus, it re-oxidizes to be available for further reduction in a redox cycle within the anode chamber in MFC. These redox mediators or electron shuttles can be molecules excreted by the bacteria in the consortium, e.g., flavins, quinones (produced by *Shewanella oneidensis*), or pigments (i.e., pyocianines produced by *Pseudomonas aeruginosa*), or they could be an inorganic species, e.g., sulfide ($S^{2-}$), which can

be produced in a sulfate-reducing consortium and oxidized on the anode surface in a more complex MET [1]. In addition to the redox mediators that can be produced by bacteria, other compounds resulting from the interaction of the microorganisms, such as sulfide or hydrogen, can also serve the MET mechanism. On the contrary, artificial redox mediators have also been evaluated. Some examples of artificial redox mediators are methylene blue, neutral red, thionine, indophenol, and benzyl viologen, to mention a few summarized in a recent survey on electroactive biofilms [66]. Figure 8 shows a schematic of the electron transfer mechanisms.

These electron transfer mechanisms must be considered in the overall MFC performance, as they contribute to the charge and substrate transport in the MFC deriving substrate concentrations profiles near the electrodes, which is related to the charge transport in the system.

## 4. Mass and Charge Transport in MFC

In a MFC (Figure 6), the oxidation of substrate (acetate) on the anode produces protons ($H^+$), and the oxygen reduction reaction on the cathode produces $OH^-$. These two key reactions drive the performance of the cell and are commonly referred to as the kinetics of the MFC responsible for the charge transfer process. Proton transport to the anode surface occurs by diffusion within the biofilm and porous anode, and through a boundary diffusion layer, resulting in a concentration difference of both substrate and protons between the anode and anolyte. At the cathode, when oxygen reduction occurs, the catalyst layer of the cathode presents a different $OH^-$ concentration than that of the bulk solution in the cathode chamber. To minimize the overpotential at the anode or cathode, a constant supply of substrate and the addition of a buffer solution to regulate the $H^+$ and $OH^-$ concentrations should be considered. It has been reported that an increase of one pH unit at the cathode and a decrease of one pH unit at the anode can result in an overpotential of approximately 59 mV [69]. Therefore, substrate mass transfer at the anode and $H^+$ removal, along with adequate oxygen and $H^+/OH^-$ diffusion at the cathode, is a key factor in MFC performance. However, it has been observed that although the $H^+$ diffusion (concentration gradient) can be improved by adding a buffer solution, this approach may be impractical on a large scale. In the case of the substrate, research has shown that a porous biofilm supported on a porous anode can facilitate its diffusion. On the contrary, the cathode and anode resistances ($R_{cat}$ and $R_{an}$) are a combination of the reactions (kinetics) and diffusion processes (mass transport) occurring at each electrode [70]. At the cathode, the only reaction that contributes to the $R_{cat}$ is the oxygen reduction reaction (ORR) in a MFC current density range ($<100$ A/m$^2$), which is lower than the current density range in fuel cells (greater than 1 A/cm$^2$), where mass transport due to diffusion constraints may be limited [70]. At the anode, diffusion resistance in $H^+$ transport from the anode is a major determinant of MFC performance and the main contributor to $R_{an}$. Low $H^+$ diffusion at the anode leads to acidification of the biofilm, whereby a decrease in the activity of the bacteria catalyzing the reaction may occur to the point that the biofilm may enter the erosion phase. It is important to mention that while at the anode, $H^+$ diffusion can be improved by using a suitable buffer solution; this may not be possible on a large scale, as mentioned above, hence effective $H^+$ diffusion remains a concern in the operation of MFCs.

Mass transfer at electrodes has been improved through various approaches. This paper will briefly describe approaches to improve mass and charge transport at the anode in view of the focus on bioanodes, which have been extensively studied in MFC with respect to applications in organic removal in wastewater treatment. There are three main strategies that have been studied and reviewed in the field of MFC to improve their performance by diminishing mass transfer losses that can hinder charge transport. These strategies are: (a) maintaining a fluid flow, (b) biofilm regulation, and (c) anode structure and material and its interfacial characterization, which are discussed below.

Maintaining fluid flow. Fluidization, for example, in forced convection, increases the thickness of the diffusion boundary and causes destabilization of the laminar boundary

layer, which transforms into a turbulent boundary layer. This turbulent boundary layer favors mass transfer near the electrode surface; therefore, turbulent flow is recommended to decrease mass transfer limitations between the anode and the bulk solution or anolyte. There are some anode designs that have been used to support the fluidization strategy. For example, helical flow pattern anode, spiral anode, rotating carbon brush anode [71], and a design that includes 3D flow through anode [72,73]. It has been observed that a flow of 7.5 L min$^{-1}$ in a helical anode can double the power output [73]. Figure 7B shows the anodes structure most conducive to forced convection.

Biofilm regulation. A biofilm that favors mass transport of substrate must be thin and porous, e.g., decreasing acetate concentration gradients, and this favors higher current production due to efficient nutrition of the bacterial community in the biofilm. One approach to investigate the structure and composition of a conductive biofilm, i.e., a biofilm in which the resistance to charge transfer is low, is by culturing the biofilm with different external loads. Ren et al. [74] reported that in a biofilm formed in the anode of an MFC inoculated with secondary effluent from a wastewater treatment plant, the strategy of applying high external resistances (1000–5000 Ω) promoted the growth of mostly filamentous bacteria, while rod-shaped bacteria grew and formed a denser biofilm at lower external resistances (i.e., 10, 50, and 250 Ω). It was observed that charge transfer resistance at the anode decreased when the biofilm was grown with low external resistance. At the same time, it was reported that in this MFC, this approach favored the minimization of the charge transfer resistance at the anode, but still its resistance was up to two-fold higher than the charge transfer resistance at the cathode. Similarly, Zhang et al. [75] reported consistent results regarding porosity of the biofilm grown at low external resistances. For example, more voids were observed in a biofilm grown at 50 Ω of external resistance versus the biofilms grown at 200 or 1000 Ω. The presence of more voids (porosity) increases substrate transport, which decreases resistance to charge transfer and results in increased electricity generation. At this point, it is important to mention that the early, single-layer attachment of biofilm-forming bacteria depends on the surface properties of the anode, e.g., hydrophobic or hydrophilic features, chemical functional groups, and texture. In the initial stages of biofilm formation, low energetic yields are observed, and as the biofilm matures, the electrocatalytic activity increases. Modification of the anode material in various ways that will be mentioned in this paper allows for increased bacterial adhesion and, thus, increased electron transfer rates [76].

Interfacial characterization. The Electrochemical Impedance Spectroscopy (EIS) technique is used for interfacial characterization between the biofilm and the electrodes. In EIS measurements, the experiment must exhibit three characteristics: linearity, causality, and stability. Three important parameters, such as biofilm capacitance, charge transfer resistance, diffusion in the biofilm, or in situ monitoring of biofilm integrity, are derived from these three characteristics [77]. The Kramers–Kronig (K-K) relationship is used to validate the data. However, this method is rarely performed in anodic or cathodic characterization of biofilms. Figure 9 shows two different experimental setups for a three-electrode assembly [78,79].

Depending on the experimental setup and biofilm-based nanomaterials, the theoretical approach becomes a complex system for fitting the Nyquist profile with a suitable equivalent electrical circuit. For example, Turik et al. [79] proposed an equivalent circuit, which is different compared to that of Kosimaningrum et al. [76]. On this basis, Figure 10 shows different equivalent electrical circuit models with a frequency from 100 KHz to 1 mHz. As we can see, the Nyquist profile changes if we consider the same solution resistance (i.e., 92 Ω).

There are some recommendations for performing EIS experiments on electrode materials [80] and electrode biofilm [77]. Finally, tests on MFCs are performed following the same protocol in Fuel Cells. In our group, different protocols for micro-laminar flow fuel cells are performed and fuel cells are implemented, which can be applied to MFCs [10,81,82].

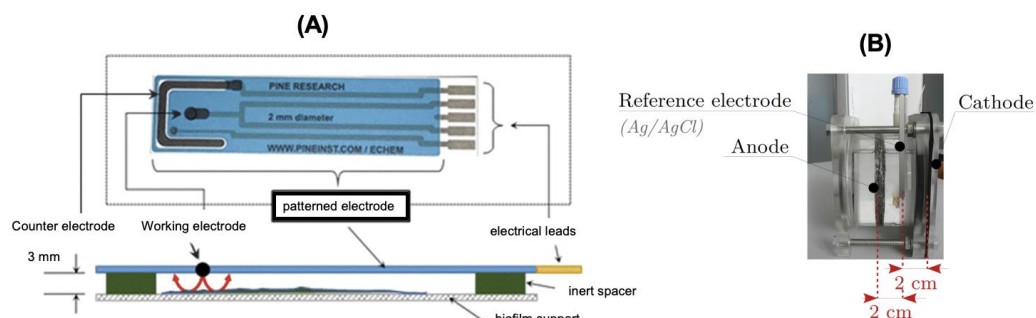

**Figure 9.** EIS experimental setups for three-electrode assembly in a flat patterned electrode (**A**) and Microbial Fuel Cell design (**B**). Figure (**A**) with permission from Ref. [79]. Copyright (2020) under Creative Commons Attribution License. Figure (**B**) with permission from Ref. [78]. Copyright (2021) IOP Publishing.

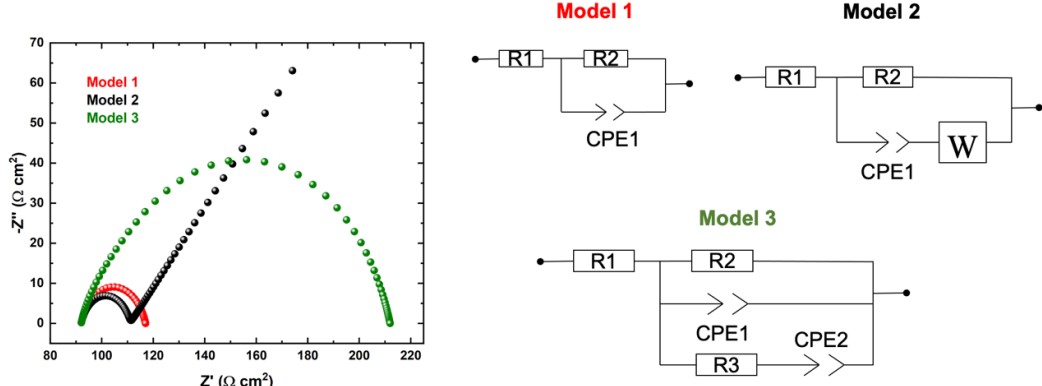

**Figure 10.** Simulation of Nyquist-EIS profiles with **Model 1:** R1 = 92 Ω, R2 = 25 Ω, Q = 20 μF, and a = 0.8; **Model 2:** R1 = 92 Ω, R2 = 19 Ω, Q = 20 μF, a = 0.8, and $A_w$ = 5 Ω s$^{-1/2}$; **Model 3:** R1 = 92 Ω, R2 = 120 Ω, R3 = 80 Ω, Q1 = 20 m μF, Q2 = 10 μF, and $\alpha_1$, $\alpha_2$ = 0.8. The geometric area was equal to 1 cm$^2$ and frequency from 100 kHz to 1 mHz.

Anode's structure and material. The anode structure and material are also a key factor in improving mass transport and decreasing charge transfer resistance due to low substrate diffusion in the biofilm. Regarding anode structure and material, research has demonstrated that a three-dimensional (3D) anode prepared with a material that can provide a high porosity favors biofilm growth and improves mass transport, which contributes to better performance of MFCs. As a result of the comprehensive reviews on structure and materials for anodes that can improve mass transport and, thus, support charge transfer resistance to decrease [83,84], some highlights can be recovered. In addition to conventional carbon-based anode materials, i.e., carbon rod, fiber, cloth, brush, foams and graphite plates, granules, rods, and activated carbon, modern anode materials have been used in MFC research. These modern materials have attracted attention due to the low cost of the raw material required for their preparation and can be classified as follows: biomass-based natural carbon, composite-based anode material, and graphene-based anode material [84]. For example, coffee waste has been reported to be an excellent material to prepare anodes due to its great conductivity and porosity [85]. Some of the natural biomass sources for making anodes by carbonization or carbonization-polymerization are bamboo charcoal, which has a power density of 1652 mW/m$^2$ in the MFC, where it was evaluated [86]; municipal sludge (power density of 568.5 mW/m$^2$); sewage sludge (power density of 2228 mW/m$^2$) [87,88]; cotton textile (power density of 931 mW/m$^2$) [12]; sugarcane (power density of 59.94 W/m$^2$) [89]; coconut shell (power density of 1069 mW/m$^2$) [90]; and corn stem (power density of 3.12 mW/cm$^2$) [91], to mention a few. The power densities for the MFCs in which these biomass-based anodes were used may be difficult to compare objectively due to the different source of microorganisms used to inoculate the

MFC, the substrate used (i.e., acetate, wastewater with a high organic load, etc.), and the operational conditions of the MFC. For example, in the case of the anode prepared with municipal sludge as feedstock, experiments to evaluate the anode were performed with *Shewanella oneidensis,* a known electrogenic bacterium, versus different consortia used in the other reports. However, in general, the power densities obtained in MFCs with these anodes prepared with natural biomass are higher than those obtained using conventional materials for anode preparation, according to the research reports.

Physicochemical characterization of electrodes

In the physicochemical characterization of electrode materials, different techniques and methods of analysis have been developed. Two blocks are described here: surface and bulk non-electrochemical techniques and surface electrochemical characterization.

Surface and Spectroscopy techniques. *Fourier Transformed infrared (FTIR) spectroscopy* is used to measure the infrared absorption or emission spectrum of a solid, liquid, or gas. However, this technique has been implemented to monitor chemical changes in biofilms attached to a substrate. In this regard, Attenuated Total Reflection (ATR)-FTIR and FTIR spectroscopy have gained a lot of attention to monitor biofilm, non-destructively, in real time, and under fully hydrated conditions [20–23,92]. Quilès et al. [93] studied a bacteria named *pseudomonas fluoresces* to understand the initial steps of biofilm formation. In addition, Holman et al. [22] developed a simple open-channel microfluidic system. They found that it is possible to directly monitor the activity of bacteria and their biochemistry at the molecular level within a biofilm for a long time. Therefore, different research groups are working to improve this method [22,94,95]. Figure 11 depicts the microfluidic membrane device schematic and the obtained results.

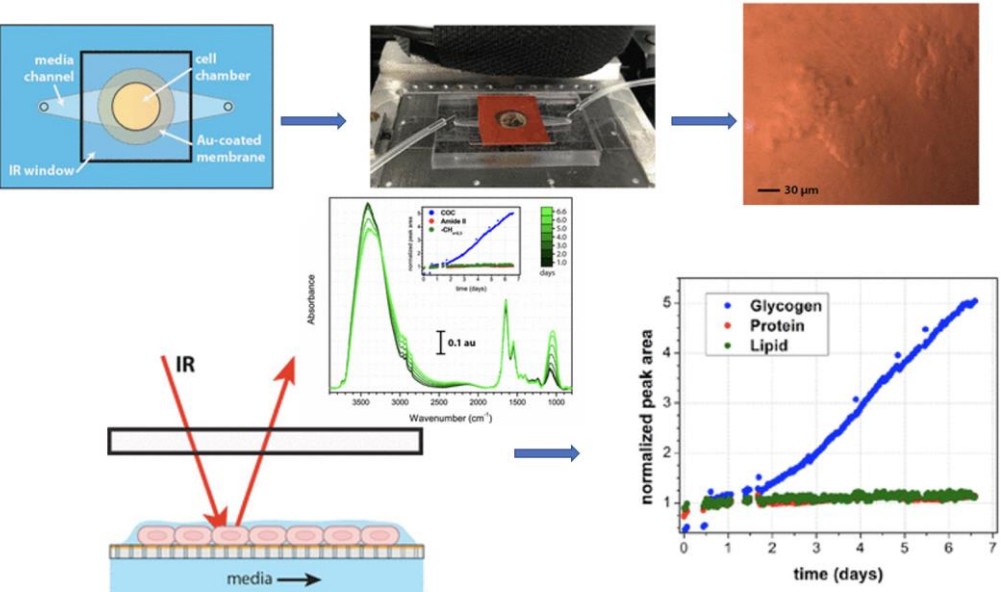

**Figure 11.** Experimental setup for in situ FTIR experiments. Figure adapted with permission from Ref. [95]. Copyright (2015) American Chemical Society.

*X-ray photoelectron spectroscopy* (XPS) is a surface spectroscopy technique that can be used to understand the cell-adhesion process. The first approach was performed by der Mei et al. [33]. They combined XPS and pH-dependent Z-potential on the cell surface, which allowed them to build a complete model of bacteria cell surface. In addition, Tayler [96] studied the importance of cell adhesion to the synthetic surface (i.e., biofilm formation). The interaction between the biofilm and the substrate induced surface corrosion. This phenomenon has been studied by some groups [97–99]. For example, Teng et al. [97] found that biofilm formation affects the compositions of different iron and calcium oxides, as well as crystalline growth [98].

Near- and far-field microscopy. In near-field microscopy, Atomic Force Microscopy (AFM) is a technique that offers a new opportunity in bacterial surface characterization, in which the cell surface topography is characterized at the nanometer scale [24,100–104]. Malki et al. [101] studied a bacterium named *Acidiphilium* sp. And biofilm formed on graphite-like substrates. According to their results, AFM images showed bacterial growth, maturation of the biofilm, excretion of extracellular polymeric substance (EPS), and the presence of nanometric particles over time. In addition, Chatterjee et al. [24] studied biofilm growth, taking into account the mode of height distribution (MPH). Furthermore, Sharma et al. [105] found that adhesion force is an important parameter to measure and quantify the interactions between cell surface materials. Meanwhile, Scanning and Transmission Electron Microscopy (SEM, TEM) allow the microstructural and morphological properties of a material to be analyzed. However, the morphology and homogeneous thickness of the biofilm are analyzed by SEM images [32,106–108]. In this case, SEM images show that the biofilm was densely packed in a multilayer structure over a period [106,109]. In addition, microorganisms can adhere to the channel walls and the electrode surface under different conditions [107]. Furthermore, a study shows the formation of nanowires between bacteria and substrate [110].

For TEM analysis, visualization of the extracellular matrix of biofilm was hampered by the technique itself. Resse and Guggenheim [28] developed a method to overcome these difficulties and used biofilms of various species grown in vitro. In this case, TEM analysis and preservation and staining methods can provide better visualization of the intracellular and extracellular features of biofilms. These findings can aid in understanding mineral nucleation in biofilm [32]. Further, Sangetha et al. [108] studied in situ *Candida albicans* biofilm, in which *Cassua sopectabilis* extract can inhibit its growth.

## 5. Structural and Mass Spectroscopy Characterization

Chen et al. [111] used the XRD technique to confirm the formation of the amorphous carbon phase before acting with bacteria as an anode. Qualitative characterization steps elucidated that graphite-cement composites (mass by weight 1:1) with 42.86 %wt. water presented the best performance applied in microbial fuel cells as a cathode [112]. Here, several research groups focus on fabricating the best materials for anodes [15,111,113,114] and cathodes [112,115–117]. However, it is possible to directly apply the XRD technique on a biofilm. Hu et al. [27] analyzed XRD data and found that amorphous phases form in biofilms attached to carriers.

Mass spectroscopy is an analytical technique to measure mainly mass-to-charge ratio (m/z) of one or several molecules present in a sample. We can distinguish in situ and ex situ forms. For ex situ forms, the sample is analyzed after the reaction has been carried out, and two modes can be used: Liquid Chromatography Mass Spectrometry (LC-MS) [29] and Ion Chromatography Mass Spectrometry (IC-MS) [30]. For example, Liu et al. [29] followed the degradation of Congo red, in which the LC-MS analysis showed the formation of small organic molecules in a microbial fuel cell. Under anoxic conditions, Wang et al. [31] demonstrated a rapid degradation from sulphamethoxazole to methane in the same microbial system.

In in situ forms, the reaction is carried out and the by-products are quantified on the same timescale. Here, we can find Electrochemical On-line Inductively Coupled Plasma Mass Spectroscopy (ICP-MS) [25] and Differential Electrochemical Mass Spectroscopy (DEMS) [26]. For the DEMS method, Kubannek et al. [26] followed the $CO_2$ production in electrochemically active bacteria during acetate oxidation. The typical components of an on-line DEMS instrument are shown in Figure 12I [118].

For ICP-MS, the experimental setup is illustrated in Figure 12II, which is appropriate to study metabolic processes of electrochemically active bacteria.

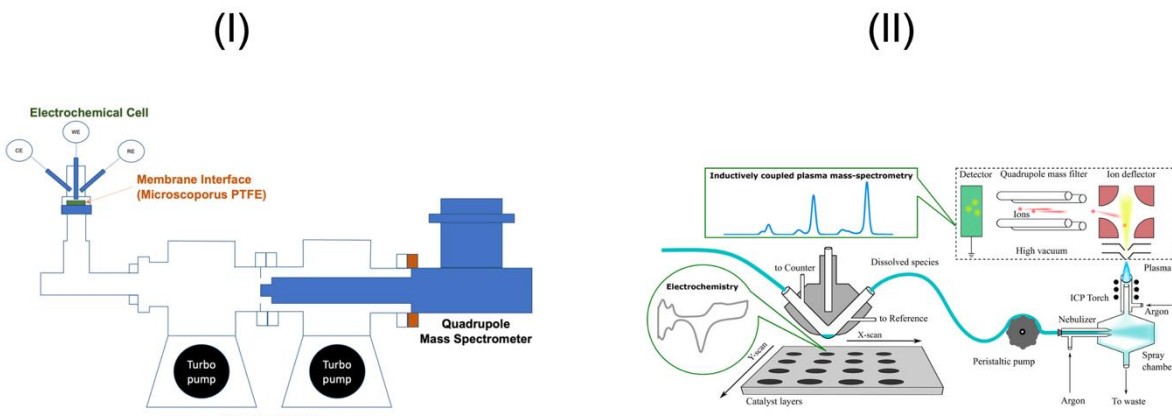

**Figure 12.** (**I**) Experimental setups for Differential Electrochemical Mass Spectroscopy and (**II**) Electrochemical On-line Inductively Coupled Plasma Mass Spectroscopy (ICP-MS). Figure (**I**) modified and adapted from Ref. [118] and Figure (**II**) with permission from Ref. [25]. Copyright (2019) John Wiley and Sons.

## 6. Surface Electrochemistry

Nowadays, electrochemical methods are simple and fast tools for the analysis of electron transfer and pathway mechanisms. Therefore, this tool has been extended to understand the mechanism of electron transfer between microbial biofilms and an electronic conductor. The most common electrochemical methods are cyclic and linear voltammetry, where we can distinguish different redox peaks. Here, Yuan et al. [119] electrochemically characterized carbon as an anodic biofilm with two different substrates: glucose and acetate. These findings are described in Figure 13A, where the current density as a function of scan rate, Figure 13B,C, on the same substrate is analyzed. According to this, the linear trend of the current density is governed by typical mass-diffusion phenomena.

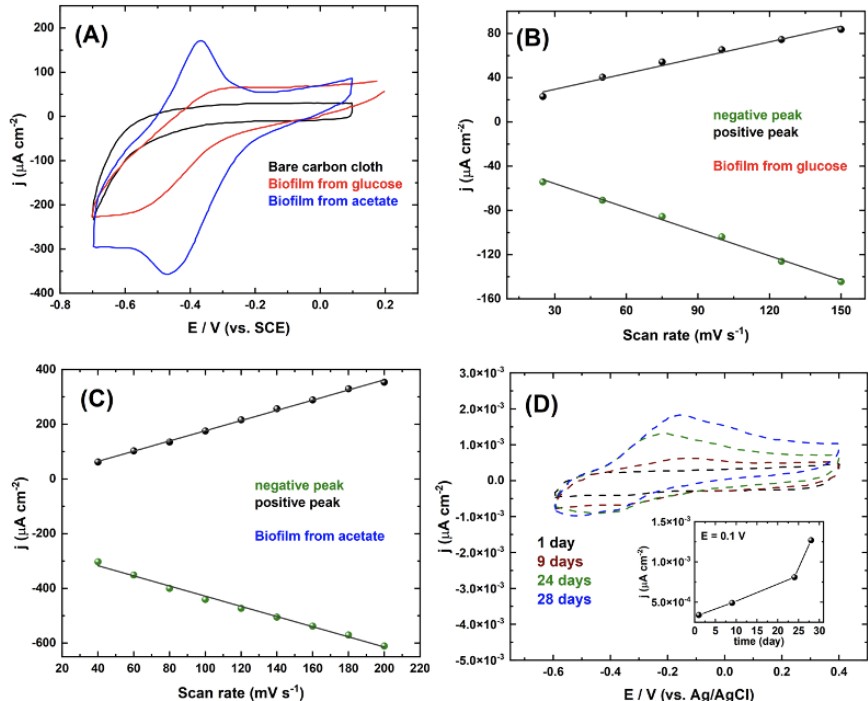

**Figure 13.** Voltammetry cyclic and density current as a function of scan rate for a biofilm with two different substrates (**A**–**C**). Adapted from Ref. [119] with permission. Copyright (2011) Elsevier. (**D**) Monitoring of MFC over time at a scan rate of 100 mV s$^{-1}$. Adapted from Ref. [120] with permission. Copyright (2013) Springer Nature.

On the contrary, Martin et al. [120] analyzed the evolution of anode cyclic voltammo-grams over a certain period of time in a MFC assembling, see Figure 13D. As can be seen, the positive density current peak increases with time, which is related to the proliferation of the anode-reducing microorganism. Chronoamperometric measurements are also performed to check the nucleation and growth of the biofilm over time [78]. Table 1 summarizes the different materials used as anode and their characterization by cyclic voltammetry.

Microbial fuel cell applications.

In this section, we reviewed some zero-, one-, two-, and three-dimensional biofilm electrode materials (0, 1, 2, and 3D-BE) that are used for energy production and water treatment processes.

0, 1, and 2D Biofilm Electrodes applied in MFC as Energy sources. Composite-based anodes have been designed to improve anode efficiency due to the synergy of two or more different materials [84]. These composites can be metal-metal oxides, conductive polymers, and combined carbon-based composites. The metal-oxide-carbon composite has been widely studied due to decreased corrosion and improved conductivity. Titanium oxide ($TiO_2$) is one of the most used metal oxide compounds to prepare metal oxide-carbon-based composites in a variety of 3D anodes, but there are other metals that have also been proven to be a viable option for anode composites and that are biocompatible. Biocompatibility has also been demonstrated for metals such as Au, Ag, Cu, Ni, Co, and stainless steel, in addition to titanium. Cu showed the best results in terms of biocompatibility with electrogenic bacteria [121]. In the case of carbon nanotubes (CNTs), some reports have indicated that they present toxicity towards the bacteria in the biofilm [122]; however, the formation of hybrid CNT-biofilm, as in multi-walled CNTs (MWCNTs), resulted in higher biofilm conductivity and improved substrate mass transport at the anode [123]. On the contrary, at a microscale (25 mL working volume double chamber MFC), the "spin spray layer by layer" (SSLbL) CNT anode proved to be a good approach for biofilm formation and power generation (0.83 W/m$^2$). This was studied with *Geobacter* sp. as inoculum, and the authors of that research concluded that the biocompatibility was improved with this SSLbL CNT anode versus a bare gold anode [124].

Anodes have also been prepared with the 2D crystalline carbon allotrope Graphene (Gr′) due to its conductivity, biocompatibility, and surface area. Composite anodes with graphene and metal oxides or conductive polymers (i.e., polyaniline) have shown improved MFCs. Sun et al. [125] presented an improvement in the power density produced (381 mW/m$^2$) in a MFC, in which a Gr′–PANI anode was used in the presence of *Shewanella oneidensis*, a known electrogenic bacteria. Despite the satisfactory results observed when using Gr′-based anodes, research focused on improving the properties of these anodes has led to the preparation of 3D Graphene-based anodes to increase the conductivity, surface area, and electrocatalytic activity [126]. In the same vein, Graphene-metal composite anodes (i.e., Graphene-CuO, Graphene-$TiO_2$, Graphene-ZnO, Graphene-AgO) are another approach that has presented reliable results in MFCs. The synthesis of these graphene-based anodes, particularly if the graphene is prepared from natural biomass material, can be explored for applications in MFCs [84].

Table 1 lists some of the anodes (anode materials) that have been shown to improve the performance of MFCs by reducing the charge transfer resistance and power densities obtained in each report. In most cases, the anode preparation is reported, along with electrochemical characterization, particularly to investigate charge transfer resistance reduction, for which the electrochemical impedance spectroscopy (EIS) technique is used. It is important to note that some reports conduct the evaluation of the anode once the biofilm has formed, and other reports include the evaluation of bare anodes, just prior to the use of the anodes in the MFCs. This approach allows appreciating the difference between the charge transfer resistances of a carbon-based anode and a composite anode of any type. The evaluation of charge transfer resistances at the anodes, once the biofilm has formed, provides further information on the role of the biocatalyst (bacteria), and the overall system performance (MFC) is also dependent on the electrogenic character of the

bacteria, whether pure strain or a consortium. According to the literature review, it is useful to estimate the charge transfer resistance of bare anodes to compare the materials and choose the one that exhibits the lowest resistance, and then, once the electrogenic bacteria have formed a biofilm, compare the charge transfer resistances again to gain information on the bacteria–anode material interaction. As shown in Table 1, the variation of power densities depends on several factors, such as the anode material (also cathode material), the bacteria used, and the MFC arrangement; therefore, each system should be designed considering the MFC application with the specific choice or anode materials.

**Table 1.** Power densities obtained with modern anode materials in MFCs.

| Anode Material | Power Density | Characterization Method to Obtain Ch TR * | Reference |
|---|---|---|---|
| Natural biomass-based material | | | |
| Bamboo charcoal | $1652$ mW m$^{-2}$ | EIS Ch TR $\downarrow$1 | [86] |
| Municipal sludge | $568.5$ mW m$^{-2}$ | EIS Ch TR $\downarrow$2 | [87] |
| Sewage sludge | $2228$ mW m$^{-2}$ | ACI Ch TR $\downarrow$3 | [88] |
| Cotton textile | $931$ mW m$^{-2}$ | EIS Ch TR $\downarrow$4 | [12] |
| Sugar cane | $59.94$ W m$^{-3}$ | | [89] |
| Coconut shell | $1069$ mW m$^{-2}$ | | [90] |
| Corn stem | ** $3.12$ mA cm$^{-2}$ | EIS Ch TR $\downarrow$5 | [91] |
| Composite materials | | | |
| Ti Graphene/PANI | $124$ mW m$^{-2}$ | | [127] |
| Fe-N-S-doped carbon tubes | $479$ W m$^{-3}$ | | [128] |
| Nano Fe$_3$C/PGC | $1856$ W m$^{-2}$ | EIS ChT TR $\downarrow$6 | [129] |
| Carbon felt/PANI | $216$ mW m$^{-2}$ | EIS ChT TR $\downarrow$7 | [130] |
| Activated carbon/PANI | $273$ mW m$^{-2}$ | EIS ChT TR $\downarrow$8 | [131] |
| MnO$_2$/carbon felt | $3580$ mW m$^{-2}$ | SC and SR | [132] |
| Stainless steel mess/PANI | $48$ mW m$^{-2}$ | EIS ChT TR $\downarrow$9 | [133] |
| Stainless steel mess/PANI/carbon nanotube | $38$ mW m$^{-2}$ | EIS ChT TR $\downarrow$10 | [133] |
| Fe carbon cloth | $890$ mW m$^{-2}$ | EIS with biofilm | [134] |
| Graphene modified based anodes *** | | | |
| Graphite block anode (and Graphite/polyester Ni as cathode) | $1575$ mW m$^{-2}$ | CV, EIS | [135] |
| Graphite/polyester Fe (graphite block cathode) | $58.92 \pm 11.27$ mW m$^{-2}$ | CV, EIS low Ch TR | [135] |
| Graphite/polyester Co (graphite block cathode) | $430$–$630$ mW m$^{-2}$ | CV, EIS low Ch TR | [135] |
| Graphite/polyester Ni (graphite block cathode) | $430$–$630$ mW m$^{-2}$ | CV, EIS low Ch TR | [135] |
| Graphite/polyester Cu (graphite block cathode) | $430$–$630$ mW m$^{-2}$ | CV, EIS | [135] |
| Graphite/polyester Zn (graphite block cathode) | $1188$ mW m$^{-2}$ | CV, EIS low Ch TR | [135] |
| Graphite/polyester Mn (graphite block cathode) | $1200$ mW m$^{-2}$ | CV, EIS lowest Ch TR | [135] |
| Graphite/polyester Mg (graphite block cathode) | $1.36 \pm 1.3$ mW m$^{-2}$ | CV, EIS | [135] |
| rGO/polydopamine | $988$ mW m$^{-2}$ | EIS Ch TR $\downarrow$11 | [29] |
| Eucalyptus leaves/rGO/carbon fiber paper | $1158$ mW m$^{-2}$ | EIS with inoculum | [136] |
| Cellulose derived GO/PANI | $1.1$ mW m$^{-2}$ | EIS with inoculum | [137] |
| Lignin derived GO | $0.148$ mW m$^{-2}$ | CV and EIS | [138] |
| Lignin derived GO/ZnO | $1.15$ mW m$^{-2}$ | CV and EIS | [138] |
| Lignin derived GO/TiO$_2$ | $0.75$ mW m$^{-2}$ | CV and EIS | [138] |

(rGO) reduced graphene oxide, (PANI) polyaniline, (Nano Fe$_3$C/PGC) iron carbide nanoparticles dispersed in porous graphitized carbon, (EIS) electrochemical impedance spectroscopy, (CV) cyclic voltammetry, (ACI), AC impedance, * Ch TR charge transfer resistance, $\downarrow$1 means lower than graphite tube, $\downarrow$2 lower than graphite felt, $\downarrow$3 lower when the anode is prepared at higher temperature (1200 °C) against graphite plates, $\downarrow$4 lower than carbon felt, Ch TR $\downarrow$5 lower than graphite electrode ** current density reported, $\downarrow$6 lower than porous graphitized carbon and carbon felt, $\downarrow$7 lower than carbon felt, $\downarrow$8, lower than activated carbon, $\downarrow$9 lower than stainless steel mess, $\downarrow$10 lower than stainless steel mess/PANI and stainless steel mess, $\downarrow$11 lower than carbon cloth. *** Graphite/polyester Mn (graphite block cathode) presented the lowest Ch TR out of all the anodes combined with metals [135], SC specific capacitance, SR specific resistance.

On the contrary, in a recent comprehensive review on microbial fuel cells configuration with electrode and membrane materials, a summary was presented highlighting several reports in which dual- and single-chamber MFC configuration were predominant. It is for these two types of configurations, for which most of the data can be found, that it is useful to obtain information on the combination of factors; however, a fair comparison between all of them is difficult. For example, although mixed cultures predominate in most cases, electrode materials vary from simple graphite and carbon felt to metal composite electrodes and membranes, i.e., Ultrex and Nafion is also used along with a selection of substrates, such as glucose, domestic and synthetic wastewater, and landfill leachate. The reported power densities vary in a wide range, with values from 5.04 to 3600 mW m$^{-2}$ [139]. According to the reviewed papers, the authors suggested that all the collected information points to dual-chamber MFC, carbon-based electrodes, carbon brush anodes, and nanomaterials as anodes as the features that most support better MFC performance, considering that in most cases, mixed cultures are used as biocatalysts.

2D and 3D Biofilm Electrodes in MFC applied to wastewater treatments. A recent literature survey conducted by one of the leaders in MFC technology [140] presented interesting data on the current status of MFC applications. According to that report, MFC research can be considered of paramount importance in water treatment, along with energy recovery, which makes this technology worthy of further expansion. In recent years (i.e., 2010–2020), evidence of MFC research has been steadily increasing, but the focus on large-scale application of this technology remains low–less than 0.5 % of all MFC reports focus on large-scale applications and commercialization [140]. Laboratory-scale studies of MFC continue to focus on improved designs, including dimensions, location, materials and number of electrodes, long-term biofilms enhancement, and appropriate analytical and electrochemical techniques to evaluate system performance. This technology requires multidisciplinary work and a thorough knowledge of the system. For wastewater treatment, it has been observed that the removal of organic matter can be effective, but the coulombic efficiency of the system can be low. Treatment of textile, distillery, and food industry water streams with MFC has been documented in layouts and designs that allow the modification of the bioreactors to a MFC configuration—for example, up flow anaerobic sludge blanket reactors (UASB 2.2 L volume) modified to a MFC configuration to treat molasses wastewater producing 1410.20 mW m$^{-2}$ [141]. On the contrary, at larger scale (720 L), a septic tank adapted to an MFC configuration has been used for on-site sanitation [34]. Another example of large-scale MFC was reported by Jadhav et al. [140] with 1500 L capacity ("Bioelectric toilet MFC" composed of 49 MFCs). To harness the energy produced on a large scale when using stacked MFCs, an effective energy management system is necessary, or the purpose of the facility would only meet the water treatment requirement, because in a stacked MFC array, the power obtained is low. It has been suggested to integrate a MFC as a part of the wastewater treatment process by, for example, inserting it as a secondary treatment unit [140]. Constructed wetlands (CW) integrating to an MFC technology is a recent approach (first report in 2012) that has gained much attention from researchers in the field [142]. A CW-MFC offers several advantages to wastewater treatment due to the power recovery (0.175–55.5 mW m$^{-2}$), along with the removal of COD, nitrogen, and in some cases, metals [143]. Several physical and biological phenomena occur in a CW-MFC, and the system is still under study, as in addition to understanding and implementing the best arrangement for a MFC in such a scenario, the effect of the macrophytes-associated microbial community and the mass and charge transport mechanisms poses challenges to the long-term and large-scale application of this bioelectrochemical system.

Electrochemical processes have high efficiencies and low residence times for the treatment of recalcitrant organic compounds. However, they require high energy consumption due to the use of electric current, and the electrodes tend to degrade due to numerous factors that shorten their lifetime, which increases the operating and investment cost of this type of technology. Bioelectrochemical systems (BES) have been conceived to synergistically

combine the advantages of electrochemical and biological processes. However, their long hydraulic retention times due to mass transfer problems in the BES have promoted the need to think about other types of devices that allow improving the primitive versions of BES. Three-dimensional biofilm electrodes (3D-BE) have been designed based on conventional bioelectrochemical systems (BES) involved in the treatment of wastewater containing refractory compounds (antibiotics, heavy metals, and dyes, among others). The 3D-BEs consist of a cathode and an anode, whose internal part is filled with conductive particles (granulated activated carbon) [19], providing a large specific area for microbial adhesion and subsequent contact with contaminants, which not only improves mass transfer, but also migration (microbial carriers). Micro-charged electrodes are formed once polarization is applied to the filled particles, thus improving the current efficiency of this type of ESP [144]. Reactors containing 3D-BE efficiently break down refractory contaminants into biodegradable compounds by electrocatalysis, which are subsequently removed by microorganisms attached to the electrode particles. The application of the electric field not only provides the promoting force to carry out the electrochemical reactions, but also stimulates microbial metabolism, allowing its development and increasing the efficiency of the biological process [145]. This combination of technologies allows increasing the tolerance of microorganisms in toxic environments with a high concentration of contaminants and ions, as well as high toxicity [134]. Micro-charged electrodes are formed once a polarization is applied on the filled particles, thus improving the current efficiency of this type of ESP [144]. In particular, the assembly of the particles in the electrodes is of special interest, as this is the key factor in increasing efficiencies, compared to other BE. However, the above has not been addressed in any study, and therefore, it is reviewed below.

As described in Table 2, the features (mechanical and electrochemical stability, ionic and electronic conductivity, catalytic activity, porosity, active area, availability, and low cost) of the 3D-BE determine the overall performance of the application. Once the voltage is applied on the electrode particles, several microelectrodes are formed, in which the contact between microorganisms and contaminants is improved, as well as the mass transfer in the reactors, compared to classical BES. In addition to participating in electrochemical reactions, the electrode particles also act as microbial carriers for colonizing organisms that promote their growth and reproduction. Within these particles, complex processes arise, such as reduction and oxidation reactions, adsorption, and electrocoagulation, among others. These combined processes increase the efficiency of the wastewater treatment process. Currently, the most used materials to manufacture 3D-BE are carbonaceous, such as graphite felts, granular activated carbon, graphite, and composite materials with metals, metal oxides and conductive polymers. Granular graphite has been used for the construction of this type of electrode due to its good biocompatibility and conductivity, although low mass transfer, compared to granulated activated carbon. Li et al. [146] fabricated an extended cathode with $FeOOH/TiO_2$/activated carbon to remove tetracycline hydrochloride antibiotics, where 3D-BE showed good catalytic activity, but poor conductivity. Cui et al. achieved good decolorization of wastewater using granular graphite electrode particles loaded with catalytic elements [147] and fabricated an extended cathode with $FeOOH/TiO_2$/activated carbon to eliminate tetracycline hydrochloride antibiotics, where 3D-BE showed good catalytic activity, but poor conductivity [146]. Guo et al. removed 95.7% dexamethasone in a 3DBE system composed of polyaniline and activated carbon, highlighting that the polymer improved the electrocatalytic activity, stability, surface area, and conductivity of the activated carbon electrode particles, compared to removals of 81.6 and 56.7% in 2D-BE and a conventional biological system, respectively [148]. Zhang and Liu improved the conductivity, porosity, and catalytic activity of $MnO_2/TiO_2$/g-$C_3N_4$/activated carbon, compared to activated carbon, during real coking wastewater treatment [149]. On the contrary, granular activated carbon has a large specific surface area and porous structure, which can be beneficial to increase the efficiency of 3D-BE. Liu et al. [150] developed a three-dimensional electrode with this material to remove 90% brilliant red X-3B (1000 mg L$^{-1}$) from wastewater and remove 80%

COD in 24 h. Similarly, Hao et al. [151] manufactured 3D-BE with granular activated carbon with a C/N ratio of 3 for removal of 98.3% of nitrates within 7 h.

**Table 2.** Parameters involved in relevant performance of particle electrode materials made with granular activated carbon in 3D-BE systems.

| Operating Conditions | Wastewater Component | Removal Percentage (%) | Reference |
|---|---|---|---|
| 120–450 mg L$^{-1}$ COD, 40–80 mg L$^{-1}$ NH$_4$$^+$$-$N, 43.92 h, and pH 7–8 | Numerous components | 95–99 | [152] |
| 200–800 mg L$^{-1}$ COD, 1.25 h, and pH 6.8–7.1 | Synthetic wastewater | 97 | [153] |
| 700 mg L$^{-1}$ COD, 0.085 mg L$^{-1}$ sulfide, pH 7 | Spent caustic | 97.56 | [154] |
| 2 mg L$^{-1}$, 0.8 V, and 36 h | Ciprofloxacin and sulfadiazine | 98.3 and 99.8 | [155] |
| 0.2 mg L$^{-1}$, 0.8 V, and 6 h | Sulfamethoxazole and tetracycline | 93.5 and 95.6 | [156] |
| 50 mg L$^{-1}$, 2 V, and 48 h | Methyl red | 89.3 | [157] |
| 1000 mg L$^{-1}$, 2.5 V, and 24 h | Reactive brilliant red X-3B | >90 | [150] |
| 30 mg L$^{-1}$, 40 mA, C/N ratio 3, 7 h, and pH 7 | NO$_3$$^-$-N | 98.3 | [151] |

In recent years, some particulate electrodes made from steel and lithium slags, ceramsite, resins, and zeolites have been used for wastewater treatment. To this regard, Feng et al. [158] used sulfonated cation-exchange resins to remove heavy metals from electroplating processes, together with an electrocatalytic step. Steel converter slag was also used for this type of electrode to remove 91.68, 91.54, and 87.63% decolorization, NH$_4$$^+$, and COD, respectively, during RhB removal [159]. Other steel and lithium slag composites were also used for wastewater treatment, showing good catalytic activity. Li et al. [160] developed a 3D-BE using lithium slag to abate salicylic acid, achieving removal efficiencies of 87.1 and 85.2% of COD and pollutant, respectively. The removal process was carried out synergistically between electrochemical oxidation mechanisms and biological and physical processes [161]. Using a similar material, Li et al. [160] compared the destruction of tetra-bromo-bisphenol A using two 3D-BEs made of zeolite and granular activated carbon, where carbon obtained better results due to its higher specific surface area. For these reasons, granular activated carbon is the most widely used material in 3D-BE due to its large specific area, high conductivity, porosity, and adequate biocompatibility.

The selection of the anode of a 3D-BE must consider the type of contaminant and the degree of its degradation because different materials with different characteristics have been used. In the case of Ti-based materials, the oxygen evolution that promotes degradation is generated by a suitable aerobic environment [162]. On the contrary, the use of graphite produces CO$_2$ that can buffer the pH of the system and provide carbon compounds to favor the growth of microorganisms [163]. In addition, graphite provides good electrical conductivity and is cost competitive, but it can degrade rapidly and produce aromatic compounds. Although Ti eliminates these drawbacks, its high price limits its application at the macroscopic level. An alternative is the development of composite materials combining carbon and metals. In this regard, Cui et al. [164] removed isophthalonitrile on an anode, made of Ti sponge and FeO, within an anaerobic fluidized bed. Feng et al. [144] used IrO$_2$ and RuO$_2$-coated Ti plate to remove RhB.

In the case of cathodes for 3D-BE, many materials with different characteristics have been used, but mainly focused on reductive degradation. Stainless steel has good mechanical stability, but forms hardly any interaction with microorganisms. An alternative has been to design stainless steel or Ti meshes to facilitate binding and growth of the consortia. Tang et al. [165] used a stainless-steel mesh as cathode in the removal of sulfates and nitrates in the range of 29.35 and 88.49%, respectively. Activated carbon fibers have low catalytic activity and stability, although they have high biological affinity, good conductivity, and large surface area. Wang et al. [166] constructed a carbon fiber felt cathode to remove 84.62% of total nitrogen from wastewater. The excessive cost of the materials involved in

the manufacture of cathodes and anodes requires the development of low-cost materials with high catalytic activity and stability.

## 7. Conclusions

In recent years, microbial fuel cells have attracted interest in applications related to treatment. In addition to their low costs, they are highly sustainable due to the wastewater generation of electrical current from organic matter waste. Some of their main components are the electrode materials, which are responsible for carrying out the electrochemical reactions for optimum performance of the entire cell, including the oxidation of biodegradable matter in microorganisms adhered to their surfaces. Here, we reviewed the electrode preparation processes, as well as the analytical techniques commonly used to characterize them, with particular emphasis on rationalizing the crucial phenomena affecting these procedures and favoring the rate of electron transfer, while exploiting novel materials enhancing the interaction between microorganisms and electrodes. It is expected that in the coming years, we will continue focusing on different aspects, as mentioned below.

- nD nanomaterials. The synthesis, stability, cost, and use of n-dimensional electrodes and combinations of nD nanomaterials provide a larger specific area for microbial adhesion and subsequent contact with contaminants. This will increase the performance to apply in toxic wastewater treatment because of numerous outstanding features (e.g., increase of the tolerance of the microorganisms in toxic environments, adequate transport phenomena).
- Electrochemical characterization. The combination of different electrochemical techniques (in situ or online) can provide a complete picture of the transfer electron mechanism for each applied biofilm electrode material. The EIS experiment must be carefully revised to establish a suitable equivalent electrical circuit.
- Physicochemical characterization. The microstructural properties of the material are important for understanding its stability, selectivity, and biocompatibility with biofilm. This analysis could be applied in in situ and ex situ mode.

**Author Contributions:** Writing—original draft preparation and conceptualization, L.A.E.-W., C.G.-B., J.V.-A.; writing-review and editing, N.A.-V. All authors have read and agreed to the published version of the manuscript.

**Funding:** This work was partially funded by Instituto Politécnico Nacional (Reference No SIP-20220825, SIP-20211456, SIP-20226979, SIP-20221207 and SIP-20221270), and Consejo Nacional de Ciencia y Tecnología (CONACyT), Reference No. 320252.

**Acknowledgments:** N. Alonso-Vante acknowledges financial support from the European Union (ERDF) 'Région Nouvelle Aquitaine'. Luis A. Estudillo-Wong, Jorge Vázquez-Arenas and Claudia Guerrero-Barajas thanks Instituto Politécnico Nacional for the financial support through the SIP-projects.

**Conflicts of Interest:** The authors declare no conflict of interest. The founding sponsors had no role in the design of the work.

## Abbreviations

The following abbreviations are used in this manuscript:

| | |
|---|---|
| ATR | Attenuated Total Reflection |
| AFM | Atomic Force Microscopy |
| BE | Biofilm Electrodes |
| BES | Bioelectrochemical systems |
| cAMP | Cyclic AMP |
| COD | Chemical Oxygen Demand |
| CVD | Chemical vapor deposition |
| CW | Constructed wetlands |
| DEMS | Differential Electrochemical Mass Spectroscopy |

| DET | Direct electron transfer |
|---|---|
| EIS | Electrochemical Impedance Spectroscopy |
| EPS | Exopolysaccharides or extracellular polymeric substance |
| FTIR | Fourier Transformed infrared |
| Gr' | 2D crystalline carbon allotrope Graphene |
| HRTEM | High-Resolution Transmission Microscopy |
| IC-MS | ion Chromatography Mass Spectrometry |
| IDET | Indirect electron transfer |
| K-K | Kramers–Kronig |
| LC-MS | Liquid Chromatography Mass Spectrometry |
| LPS | Lipopolysaccharides |
| MET | mediated electron transfer |
| MFC | Microbial fuel cell |
| MCP | methyl accepting chemotaxis protein |
| MWCNT | multi-walled CNTs |
| ORR | Oxygen Reduction Reaction |
| PGM | Platinum group metals |
| SDG | Sustainable Development Goals |
| SEM | Scanning Electron Microscopy |
| SSLbL | spin spray layer by layer |
| XRD | X-ray Diffraction |
| XPS | X-ray photoelectron spectroscopy |

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
