# Peer review of "Revisiting Current Trends in Electrode Assembly and Characterization Methodologies for Biofilm Applications"

_surfaces, doi:10.3390/surfaces6010002_

Round 1

Reviewer 1 Report

General comments

In this manuscript, the authors try to critically revisit the synthesis of nD nanomaterials (n = 0, 1, 2 and 3) of particular interest in MFCs, the assembly methods of a biofilm-based electrode material, the in-situ and ex-situ physicochemical, the electrochemical characterizations of materials, and the phenomena controlling the electron transfer mechanisms. These could identify the steps that determine the rate of electron transfer, while exploiting novel materials that enhance the interaction that arises between microorganisms and electrodes. This topic is very interesting, however, there are still some drawbacks need to be revised in the manuscript before publication.

Specific comments

1. For the abstract part, there are too many sentences to talk about background (lines15-22). Please focus on the critical content of this manuscript in the abstract part. Also explain the reasons of applying nD nanomaterials (n = 0, 1, 2 and 3).

2. For the introduction part, more references with detail data should be added in the manuscript. The effects of introduced nD nanomaterials (n = 0, 1, 2 and 3), in-situ and ex-situ  approaches and electron transfer mechanism should all be clarified.

3. For the electrode preparation methods part, the effects of applying nD nanomaterials should be clarified with detail data in the manuscript.

4. For the electron transfer mechanisms part, DET and IDET are both common knowledge for readers. What is the novelty or state of the art of this part?

5. The part of future work should be summarized and predicted for every individual point mentioned in this manuscript.

6. For the “conclusion” part, the significance and novelty of the current study should be strengthened here.

Author Response

(Reviewer 1)

In this manuscript, the authors try to critically revisit the synthesis of nD nanomaterials (n = 0, 1, 2 and 3) of particular interest in MFCs, the assembly methods of a biofilm-based electrode material, the in-situ and ex-situ physicochemical, the electrochemical characterizations of materials, and the phenomena controlling the electron transfer mechanisms. These could identify the steps that determine the rate of electron transfer, while exploiting novel materials that enhance the interaction that arises between microorganisms and electrodes. This topic is very interesting, however, there are still some drawbacks need to be revised in the manuscript before publication.

Specific comments

  1. For the “abstract” part, there are too many sentences to talk about background (lines15-22). Please focus on the critical content of this manuscript in the “abstract” Also explain the reasons of applying nD nanomaterials (n = 0, 1, 2 and 3).

Answer: The authors agree with the reviewer concerning the background content. Some of these lines have been removed from the abstract to make it concise. The reasons to apply nD nanomaterials are situated at the end of the abstract (page 1, lines 20 - 27):

  1. For the “introduction” part, more references with detail data should be added in the manuscript. The effects of introduced nD nanomaterials (n = 0, 1, 2 and 3), in-situ and ex-situ approaches and electron transfer mechanism should all be clarified.

Answer: We added more reference in the introduction part (references 20 - 28 and 29 - 33, page 2 lines 62 - 68) and we added some details in the end of the introduction (page 2, lines 80 - 82).

  1. For the “electrode preparation methods” part, the effects of applying nD nanomaterials should be clarified with detail data in the manuscript.

Answer: The section 2 “electrode preparation methods” indeed focused on the synthesis of different nD materials. However, the answer to the query made by the reviewer can be found in section 6 and in the section "Microbial fuel cell applications". In the latter section we can observe this effect, both for the power density obtained (see Table 1) and for the percentage of removal achieved (see Table 2). Thus, the text gives an account of the biocompatibility of the different nD materials with microorganisms, their use as an energy source and the treatment of wastewater in environmental or synthetic samples.

  1. For the “electron transfer mechanisms” part, DET and IDET are both common knowledge for readers. What is the novelty or state of the art of this part?

Answer: The authors agree with the reviewer to some extent. DET and IDET are usually mentioned in literature reviews and experimental reports on MFC as well-known electron transfer mechanisms. A recent report [1] describes that the direct interspecies electron transfer DET is a mechanism more adaptable to changes in organic loads and in the presence of contaminants. This was observed during the oxidation of organic waste during anaerobic digestion by bacteria capable of establishing electrical connections with methane producing microorganisms through protein – based conductive circuits. It was observed that the direct interspecies electron transfer was faster than the interspecies electron exchange via diffusive electron carriers (or electron shuttles). This finding is to the best of our knowledge the most recent regarding direct electron transfer and the role that it may play in the enhancement of methane production in anaerobic digestion, which is one of the most relevant bioprocesses to generate bioenergy. A clarification was added (page 9, lines 286 - 290).

Therefore, the reason to incorporate the DET and IDET in the review is also to establish a common language with new readers in the field of MFC devices. Likewise, to establish basic definitions to avoid looking at another reference. Herein, we revised some parts and added some lines (page 8, lines 263 - 272)

  1. The part of “future work” should be summarized and predicted for every individual point mentioned in this manuscript.

Answer: We summarized some details for the future work in the section conclusions (page 22, lines 788 - 800).

  1. For the “conclusion” part, the significance and novelty of the current study should be strengthened here.

Answer: This information has been highlighted on the conclusions (page 22, lines 780 - 784):

References

  1. Zhao, Zhiqiang, Yang Li, Yaobin Zhang, and Derek R. Lovley. Sparking Anaerobic Digestion: Promoting Direct Interspecies Electron Transfer to Enhance Methane Production. iScience 2020, 23, 1 - 33; DOI: 10.1016/j.isci.2020.101794.
  2. Greenman, John, Iwona Gajda, Jiseon You, Buddhi Arjuna Mendis, Oluwatosin Obata, Grzegorz Pasternak, and Ioannis Ieropoulos. Microbial Fuel Cells and Their Electrified Biofilms. Biofilm 2021, 3, 100057-100073; DOI: 10.1016/j.bioflm.2021.100057.

Reviewer 2 Report

MFCs is a hot research field in recent years, and many researchers have carried out relevant research on it. This review summarizes the relevant papers, which has a good research significance. The paper can be accepted after revision, and detail comments as following.

1. Please further clarify the relationship between electrode material, configuration and microbial activity, as well as the relationship between electrode material, configuration and electricity generation?

2. According to the general rule of microbial growth, excessive current will kill and kill microorganisms. Please figure out the relationship between MFCs' electricity production and microbial activity?

3. Microbes have normal metabolism, and the biofilm has new and old changes. Fig. 7B cannot fully explain the rule of microbial membrane changes. This figure is similar to the biological changes in traditional bioreactors, and does not reflect the specificity of MFCs. The paper needs to be further clarified.

4. How to ensure the stability of MFCs? How to effectively regulate and control different pollutant substrates to ensure the stability of MFCs?

Author Response

(Reviewer 2)

Comments and Suggestions for Authors

MFCs is a hot research field in recent years, and many researchers have carried out relevant research on it. This review summarizes the relevant papers, which has a good research significance. The paper can be accepted after revision, and detail comments as following.

  1. Please further clarify the relationship between electrode material, configuration, and microbial activity, as well as the relationship between electrode material, configuration, and electricity generation?

Answer: The combination of factors that influence the MFCs performance may be large enough to make a fair comparison and determine a “unique” or a “most” suitable design for a MFC that includes the best combination of configuration, electrodes configuration, structure and material, microbial community etc., However, a consensus based on the observations reported in literature has been presented in some cases, for example the findings reported recently in a review prepared in 2021 [1]. A paragraph was added to the manuscript to mention some of the main findings on the combination of factors that are reflected in power density in MFCs. Cf. pages 11 and 19, lines 368-374 and 616-628.

  1. According to the general rule of microbial growth, excessive current will kill and kill microorganisms. Please figure out the relationship between MFCs' electricity production and microbial activity?

Answer: This manuscript was meant to revise the electrodes assembly and characterization in MFCs in which the current is generated “spontaneously” through the redox reactions (i.e., oxidation of organic compounds) that are carried out by the microorganisms used as the biocatalysts of the system. Since current is not applied to the MFCs and the current obtained depends largely on the microbial community grown in the biofilm attached to the anode we did not address the issue of current limits applied to electrolytic systems.

  1. Microbes have normal metabolism, and the biofilm has new and old changes. Fig. 7B cannot fully explain the rule of microbial membrane changes. This figure is similar to the biological changes in traditional bioreactors and does not reflect the specificity of MFCs. The paper needs to be further clarified.

Answer: We described a general mechanism for biofilm formation, but we mentioned what has been observed for MFC according to research reports, cf. page 7-8, lines 227 - 236.

  1. How to ensure the stability of MFCs? How to effectively regulate and control different pollutant substrates to ensure the stability of MFCs?

Answer: The authors agree with the reviewer regarding the importance of regulating the pollutant substrates to ensure stability of MFCs. However, the aim of this work was to provide a comprehensive update regarding the emerging materials and processes to fabricate the anode (or electrodes) for MFCs, which in turn contributes to highly improve the performance of this technology (MFC). Once the technology on anodes fabrication and characterization is updated and the knowledge on performance of the new developed anodes shared with the scientific community, the MFCs potential may be assessed on stability and will be able to be compared to other bioreactors.  Up to date, in view of the disadvantages that a “poor” or “low performance” anode may pose to the MFC overall performance, a comparison of this technology with other bioreactors may still be unfair and ensuring stability of MFCs at large scale on several pollutants somewhat difficult.  Therefore, the aim of this work was to provide with the recent updates on anode (electrodes) preparation processes, materials, and characterization.

Ref:

1. Greenman, John, Iwona Gajda, Jiseon You, Buddhi Arjuna Mendis, Oluwatosin Obata, Grzegorz Pasternak, and Ioannis Ieropoulos. Microbial Fuel Cells and Their Electrified Biofilms. Biofilm 2021, 3, 100057-100073; DOI: 10.1016/j.bioflm.2021.100057.

Round 2

Reviewer 1 Report

All the comments suggested by the reviewers are considered, and the manuscript was carefully modified.

Reviewer 2 Report

The manuscript has been revised, and I think it can be accepted for publication.